# Anti-Influenza Activity of Medicinal Material Extracts from Qinghai–Tibet Plateau

**DOI:** 10.3390/v14020360

**Published:** 2022-02-10

**Authors:** Olga Kurskaya, Elena Prokopyeva, Hongtao Bi, Ivan Sobolev, Tatyana Murashkina, Alexander Shestopalov, Lixin Wei, Kirill Sharshov

**Affiliations:** 1Laboratory of Molecular Epidemiology and Biodiversity of Viruses, Federal Research Center of Fundamental and Translational Medicine, 630117 Novosibirsk, Russia; sobolev_i@hotmail.com (I.S.); murashkinatatiana89@gmail.com (T.M.); shestopalov2@ngs.ru (A.S.); sharshov@yandex.ru (K.S.); 2Medical Department, Novosibirsk State University, Novosibirsk 630090, Russia; 3Qinghai Provincial Key Laboratory of Tibetan Medicine Pharmacology and Safety Evaluation, Northwest Institute of Plateau Biology, Chinese Academy of Sciences, Xining 810008, China; bihongtao@hotmail.com; 4CAS Key Laboratory of Tibetan Medicine Research, Northwest Institute of Plateau Biology, Xining 810001, China; lxwei@nwipb.cas.cn

**Keywords:** extract, *A. sativa*, *H. vulgare*, *H. rhamnoides*, *L. ruthenicum*, *N. tangutorum*, *C. rangiferina*, *A. luteo-virens*, H3N2, antiviral drugs

## Abstract

To discover sources for novel anti-influenza drugs, we evaluated the antiviral potential of nine extracts from eight medicinal plants and one mushroom (*Avena sativa* L., *Hordeum vulgare* Linn. var. nudum Hook. f., *Hippophae rhamnoides* Linn., *Lycium ruthenicum* Murr., *Nitraria tangutorum* Bobr., *Nitraria tangutorum* Bobr. by-products, *Potentilla anserina* L., *Cladina rangiferina* (L.) Nyl., and *Armillaria luteo-virens*) from the Qinghai–Tibetan plateau against the influenza A/H3N2 virus. Concentrations lower than 125 μg/mL of all extracts demonstrated no significant toxicity in MDCK cells. During screening, seven extracts (*A. sativa*, *H. vulgare*, *H. rhamnoides*, *L. ruthenicum*, *N. tangutorum*, *C. rangiferina*, and *A. luteo-virens*) exhibited antiviral activity, especially the water-soluble polysaccharide from the fruit body of the mushroom *A. luteo-virens*. These extracts significantly reduced the infectivity of the human influenza A/H3N2 virus in vitro when used at concentrations of 15.6–125 μg/mL. Two extracts (*N. tangutorum* by-products and *P. anserina*) had no A/H3N2 virus inhibitory activity. Notably, the extract obtained from the fruits of *N. tangutorum* and *N. tangutorum* by-products exhibited different anti-influenza effects. The results suggest that extracts of *A. sativa*, *H. vulgare*, *H. rhamnoides*, *L. ruthenicum*, *N. tangutorum*, *C. rangiferina*, and *A. luteo-virens* contain substances with antiviral activity, and may be promising sources of new antiviral drugs.

## 1. Introduction

Many viral infections pose a significant threat to human health, often causing death and large economic losses. Seasonal influenza affects millions of people around the world every year, causing acute contagious respiratory infections. In particular, young children, the elderly, and patients with chronic diseases are at high risk of developing severe complications due to influenza virus infection, which leads to high mortality rates [1].

Although vaccination is the primary strategy for the prevention of influenza infection, the rapid accumulation of mutations in the influenza A virus genomes enables the emerging viruses to evade the immunity developed after vaccination or previous infections with influenza A, and cause yearly epidemics and major pandemics with high morbidity and a large number of severe and fatal cases. Vaccination failures have been widely documented, and in the elderly, where most of the mortality occurs, vaccines are only approximately 50% effective [2]. In the eventuality of a pandemic infection with a new strain, antiviral drugs represent the first line of defense.

As such, the reliable prevention and treatment of influenza are critical to public health. Currently, the three main groups of official anti-influenza medicines are M2 ion channel inhibitors, neuraminidase (NA) inhibitors, and polymerase inhibitors. The M2 inhibitors (adamantanes: Amantadine^®^ and Rimantadine^®^) block the M2 ion channel that is involved in the release of NP–RNA complex in the infected cell cytoplasm. Currently, several NA inhibitors are used in clinical practice: phosphate oseltamivir (Tamiflu^®^), zanamivir (Relenza^®^), peramivir, and laninamivir. NA inhibitors inhibit the release of progeny virions from the cells and promote the clumping of virions. Polymerase inhibitors (ribavirin (Virazole, Rebetol); baloxavir marboxil (Xofluza^®^)) inhibit viral replication by targeting the endonuclease function of the viral polymerase complex [3,4].

However, influenza viruses rapidly mutate and become resistant to current antivirals [5]. Thus, essentially, all influenza strains have now developed high resistance against M2 blockers. Moreover, the efficacy of M2 blockers is limited to influenza A only, since influenza B viruses lack M2. Oseltamivir-resistant viruses (containing the H274Y NA mutation) rapidly became predominant among human seasonal A/H1N1 isolates in the 2007–2008 influenza season. Such events may lead to the widespread selection of such mutants, making the population vulnerable to a drug-resistant virus epidemic. The emergence of drug-resistant variants of the influenza virus has created the need to identify novel and effective antiviral agents. Recently, significant attention has been paid to using eco- and bio-friendly plant-based products for the prevention and cure of different human diseases. Considering the adverse effects of synthetic drugs, one approach is the screening of natural products derived from plants, identifying those with an antiviral effect, and further studying the mechanisms at the molecular level. Empirical knowledge of the ethnomedical benefits of plants, coupled with bioassay-guided fractionation and isolation, has the potential to identify novel antivirals that may be used against influenza [6].

Traditional medicinal plants have been recognized as a rich source of candidate compounds for the development of pharmaceuticals [7,8]. Many natural products and extracts from medicinal plants have been reported to possess anti-influenza virus activity [6,9,10].

Therefore, traditional medicinal plants have become an important source of candidate compounds for the discovery of novel anti-influenza drugs. Specifically, the Qinghai–Tibetan plateau, due to its high altitude and large diurnal amplitude, is referred to as the Earth’s third pole. Over 7000 indigenous medicinal plant species can be found in China, nearly 2000 species of which have been used as traditional Tibetan medicines for thousands of years [11]. Currently, more than one-third of clinical drugs have been derived from botanical extracts and/or their derivatives. Unfortunately, most medicinal plants have not been domesticated and, currently, there is no tool kit to improve their medicinal attributes to increase their clinical efficacy [12]. Thus, to discover sources for novel anti-influenza drugs, we evaluated the anti-influenza activity of nine extracts from eight medicinal plants and one mushroom from the Qinghai–Tibetan plateau, using the A/H3N2 influenza virus in Madin Darby Canine Kidney (MDCK) cell culture.

*Avena sativa* L. and *Hordeum vulgare* Linn. var. nudum Hook. f. are two Gramineae crops. In traditional Tibetan medicine, *A. sativa* is mainly used for treating pharyngeal and laryngeal diseases, whereas *H. vulgare* is used to treat lung and stomach diseases [13]. As for *Hippophae rhamnoides* Linn., *Lycium ruthenicum* Murr. and *Nitraria tangutorum* Bobr., their fruits are traditional medicinal foods consumed by Tibetans for thousands of years, and are recorded in the classic Tibetan pharmacological book *Crystal Pearl of Materia Medica* [14]. In our previous study, we found that their water-soluble polysaccharides exhibit antifatigue activity [15]. Additionally, anthocyanins from *Nitraria tangutorum* Bobr. by-products were found to have an appreciable cardioprotective effect on doxorubicin-induced injured H9c2 cardiomyocytes [16]. *Potentilla anserina* L. is widely distributed across the Qinghai–Tibetan plateau, and its roots are employed as a traditional Tibetan medicine for the treatment of malnutrition, anemia, diarrhea, and hemorrhage [17]. Recent studies showed that water-soluble polysaccharides from the roots of *P. anserine* have remarkable antioxidant and immunomodulatory activities [17,18,19]. *Cladina rangiferina* (L.) Nyl. is an indigenous lichen in the Qinghai–Tibet plateau, and its dendritic parts have been used in medicinal diets for over 500 years, often eaten as a salad after blanching to treat dysphoria with a smothery sensation, throat desiccation, sputum stagnation, blurred vision and dysopia, and jaundice [20]. Recently, we found that its polysaccharides have potential as natural antioxidants for the treatment of lung oxidative damage induced by lead in air pollutants [21]. *Armillaria luteo-virens* (Aalb.et Schw:Fr.) Sacc., mainly distributed in the meadows and grasslands of the Qinghai–Tibet plateau, is an edible mushroom and used as a traditional Tibetan medicine for the treatment of dizziness, headaches, neurasthenia, insomnia, numbness in limbs, and infantile convulsions [22]. Its polysaccharide was preliminarily found to have antioxidant and anticancer activities [23].

In this study, we investigated nine extracts obtained from the above-listed medicinal plants, lichen, and fungi for their antiviral potential against influenza virus A/H3N2.

## 2. Materials and Methods

### 2.1. Preparation of the Medicinal Plant, Lichen, and Mushroom Extracts

The fruits of *H. rhamnoides*, *L. ruthenicum*, and *N. tangutorum* were collected from Dongshangen of Dulan Country, Haixi national municipality, Mongolia and Tibet, Qinghai Province, China, in August 2011. The dendritic part of *C. rangiferina* was collected from the Qilian Mountain area in Qinghai Province, China, in July 2012. *A. sativa*, *H. vulgare*, and *P. anserina* were cultivated at Xining, Qinghai Province, China, and harvested in August 2016. The fruit bodies of *A. luteo-virens* were collected from Ebao of Qilian Country, Qinghai Province, China, in August 2016. Plant materials were identified by Prof. Xuefeng Lu, Northwest Plateau Institute of Biology, Chinese Academy of Sciences in Xining, China. The origin of extracts, full their names and abbreviations are shown in the Table 1.

ASWP and HVWP were extracted from the brans of *A. sativa* and *H. vulgare*, respectively, as described by Gangopadhyay et al. [24], and the key factors for extraction were as follows: solid-to-liquid ratio of 1:5 (*w/w*), extraction time of 4 h, temperature of 56 °C, and pH 6.6. After filtration through three layers of gauze (100 mesh), the aqueous filtrate was concentrated, then 95% ethanol was added to the aqueous filtrates up to 80% to precipitate the polysaccharides, which were collected by centrifugation and dried in a vacuum. The precipitate was dissolved in water (5% *w/v*) and the insoluble substances were removed by centrifugation. The supernatant was loaded into a DEAE-cellulose column, and then eluted with distilled water. The eluent was collected, concentrated to a small volume, dialyzed (MWCO 3500 Da), and finally lyophilized to obtain the polysaccharides.

HRWP, LRWP, and NTWP were extracted by hot water from the fruits of *H. rhamnoides*, *L. ruthenicum*, and *N. tangutorum*, respectively, following the procedure described in our previous study [15]. Anthocyanins (NJBAE) were extracted from NTWP using the SMU-AEE method and fractionated by cation-exchange resin chromatography, as described in our previous study [16].

PAWP was extracted from the root of *P. anserina* using hot water. After filtration through three layers of gauze (100 mesh), the aqueous filtrate was concentrated; subsequently, 95% ethanol was added to the aqueous filtrates up to 80% to precipitate the polysaccharides, which were collected by centrifugation and dried in a vacuum.

CRWP-S was obtained from the dendritic part of *C. rangiferina* by hot-water extraction and freeze–thawing separation, according to the procedure in our published study [23].

ALWP was extracted from the fruit bodies of *A. luteo-virens* by hot water and underwent the same steps of filtration and concentration as the PAWP.

Prior to use, all the extracts were reconstituted in PBS with 10% dimethylsulfoxide (DMSO, SIGMA) and filtered using a 0.22 μm filter.

### 2.2. Cells

The MDCK (State Research Center of Virology and Biotechnology Vector, Russia) cells were grown at 37 °C with 5% CO_2_ in Eagle MEM (Invitrogen), supplemented with 10% fetal bovine serum (FBS; Invitrogen). Before adding the extracts or the virus, the monolayer was washed twice with phosphate-buffered saline (PBS, pH 7.2–7.4 at room temperature). In all experiments, the following controls were included: cell control (cells that were not infected with the virus or treated with the extracts) and virus control (cells that were infected only with the virus but not treated with the extracts in the antiviral assays).

The total sugar content was determined by the phenol–H_2_SO_4_ method using glucose as a standard. Uronic acid content was determined by the m-hydroxydiphenyl colorimetric method using galacturonic acid as a standard [25]. Total protein and peptide contents were determined by the Coomassie brilliant blue method using bovine serum albumin as the standard [26]. The total anthocyanin content (monomeric plus condensed) of NJBAE was determined by the spectrophotometric method, and the concentration of total anthocyanins is expressed as cyanidin-3-glucoside equivalents per gram (mg CGE per g) [27]. Contaminant endotoxin was analyzed by the gel-clot *Limulus amebocyte* lysate assay and expressed as endotoxin units per milliliter (EU/mL) [28].

### 2.3. Virus

The A/Novosibirsk/RII-27192S/2020 (H3N2) strain was isolated from a human nasopharyngeal wash in March 2020. The sample was tested using real-time polymerase chain reaction with a commercial AmpliSens^®^ Influenza virus A/B-FRT kit (Interlabservis, Russia). The strain was used in an in vitro microinhibition assay and a hemagglutination inhibition (HI) test. Virus stocks were grown in MDCK cells using Eagle MEM supplemented with 0.2% bovine serum albumin (Sigma) and 2 μg/mL trypsin (Sigma) at 37 °C in 5% CO_2_ for three days. Supernatants containing the virus were collected after cytopathic effects were noted, and viral titers were determined using the 50% tissue culture infectious dose. All aliquots of virus stocks were stored at −80 °C until use.

### 2.4. Cytotoxicity Studies of Extracts

MDCK cells were seeded into 96-well, flat-bottomed microtiter plates (TPP) at 3 × 10^4^ cells/well. Following overnight incubation, the MDCK cell media were removed, and 100 µL of serial plant extract dilutions in Eagle MEM with 2 μg/mL TPCK-trypsin was added to each well (two-fold dilutions, ranging from 15.6–500 μg/mL), and another 100 μL of medium was then added to each well.

Cellular metabolic activity, as an indicator of cell viability under the influence of the extracts, was assessed using tetrazolium salt (3-(4,5-dimethylthiazol-2-yl)-2,5-diphenyltetrazolium bromide (MTT). In brief, MDCK cells were incubated for 24 h and treated in quadruplicate with serial two-fold dilutions of the extracts in maintenance media. A series of two-fold dilutions of PBS (pH 7.4) in maintenance media was used as the control. Plates were incubated at 37 °C in 5% CO_2_ for 3 days. The morphology and viability of cells were evaluated daily using light microscopy. By the 3rd day of incubation, cells were washed with sterile PBS (pH 7.4) prewarmed to 37 °C. Then, we added 10 μL of the MTT labeling reagent (final concentration 0.5 mg/mL) to each well. After 3 h of further incubation of MDCK cells at 37 °C, formazan was solubilized by adding DMSO to each well, and the optical density was measured at 540 nm using a Bio-Rad iMark TM microplate reader (Bio-Rad). The proportion of living cells in compound-treated cells relative to cell controls (defined as 100% viability) was calculated to determine the cytotoxic concentration of the extracts to cause death to 50% of viable cells in the host (CC_50_).

### 2.5. In Vitro Microinhibition Assay

We seeded 96-well plates with 3 × 10^4^ cells/well and then incubated the plates for 24 h at 37 °C with 5% CO_2_ until a confluent monolayer was attained. The MDCK cells were washed twice with PBS, and two-fold serial dilutions of extracts (7.8–125 μg/mL) in Eagle MEM were challenged with 100 TCID_50_ of the A/H3N2 virus; alternatively, cells were left untreated (negative control), treated with oseltamivir (Hoffmann-La Roche, Basel, Switzerland) ranging from 10 to 0.625 μg/mL, treated with oseltamivir and A/H3N2 virus, or treated with A/H3N2 virus (positive control). After incubation for 72 h at 37 °C and 5% CO_2_, the results were quantified using an MTT assay as previously described. The antiviral activity curve was then generated by plotting the percentages of virus inhibition against concentrations of extracts. We used the inhibitory concentration values of the extracts at which 50% of virus was neutralized (IC_50_). The IC_50_ values were calculated in Excel 2016 for Mac by linear regression analysis of percentage inhibitions. All the experiments were conducted in triplicate and data are presented as the mean ± standard deviation (mean ± SD).

### 2.6. Hemagglutination Inhibition (HI) Test

Serial two-fold dilutions of extracts (7.8–125 μg/mL) were created in 25 μL of PBS in 96-well, V-bottomed plates. Influenza viruses in PBS (25 µL/well containing 4 hemagglutinating units) were added to each dilution, and the plates were incubated for 30 min at room temperature. We added 50 µL of 1% guinea pig red blood cells (RBCs) in PBS to each well. The following controls were included in every plate: RBC without virus or extracts, RBC with virus devoid of extract, and RBC with extracts devoid of virus. The hemagglutination reactions were observed after 1 h of incubation at room temperature. The highest dilution of the virus that caused complete hemagglutination was considered the HA titration end point.

## 3. Results

### 3.1. Characteristics of Extracts Obtained from Medicinal Plants, Lichens, and Fungi

The total sugar, uronic acid, total protein and peptide, and anthocyanin contents of nine medicinal plant, lichen, and mushroom extracts were determined, and the results are shown in Table 2. Of these, eight extracts mainly contained polysaccharide components: the total sugar contents of ASWP, HVWP, HRWP, LRWP, NTWP, and CRWP-S were all above 92%. The uronic acid content assay indicated that HRWP and PAWP contained pectic polysaccharides. The total protein and peptide content assay suggested that ALWP contained a number of proteins. The total anthocyanin content assay showed that the largest quantity of this substance was contained in the NJBAE extract; a small quantity of anthocyanin was also observed in some other polysaccharide extracts, such as LRWP, NTWP, PAWP, and ALWP. Additionally, the endotoxin level in each extract solution was less than 0.5 EU.

### 3.2. Cytotoxicity Studies of Medicinal Plant, Lichen, and Mushroom Extracts

The medicinal plant, lichen, and mushroom extracts were screened for cellular toxicity in order to determine the appropriate concentrations for the in vitro microinhibition assays. The concentration associated with 50% cytotoxicity (CC_50_) was higher than 125 μg/mL for all samples (Figure 1). Therefore, serial dilutions ranging from 15.6–500 μg/mL were chosen for our in vitro microinhibition assays.

### 3.3. Inhibitory Effects of Plant, Lichen, and Mushroom Extracts on Influenza Virus

The medicinal plant, lichen, and mushroom extracts were subjected to an in vitro microinhibition screening assay to determine their antiviral activity against A/H3N2 virus (Table 2). Those demonstrating more than 50% viral inhibition were deemed to have anti-influenza activity. The MTT assay showed that the ASWP, HVWP, HRWP, LRWP, NTWP, and CRWP-S extracts notably increased the viability of MDCK cells infected with the A/H3N2 virus, compared to the virus control group, in the concentration range of 15.6–125 μg/mL. Notably, the ALWP extract also exhibited a high antiviral effect at low concentrations (7.8 μg/mL; Figure 2). The NJBAE and PAWP extracts had no inhibitory activity on strain A/Novosibirsk/RII-27192S/2020 (H3N2).

### 3.4. Inhibitory Effects of Medicinal Plant, Lichen, and Mushroom Extracts on Influenza-Virus-Induced Hemagglutination

The influenza virus HA mediates attachment to sialic acid residues expressed by the glycoproteins and glycolipids of host cells, which is a critical step in the initiation of infection. Similarly, the viral HA binds to sialic acids expressed on the surface of erythrocytes, resulting in hemagglutination. To determine whether the plant, lichen, and mushroom extracts would prevent the virus particles from binding to cell-surface receptors, we used the HI assay. Pretreatment with the studied extracts in a concentration range of 7.8–125 mg/mL could not prevent the binding of A/H3N2 virus to RBCs in this assay. These findings suggest that the studied extracts do not block the binding of viruses to cell receptors by directly interfering with viral HA.

## 4. Discussion

Phytomedicines have been known since ancient times due to their wide virucidal and bactericidal properties; however, clinical studies regarding the therapeutic potential of plant, lichen, and mushroom extracts against influenza A/H3N2 viral diseases are limited.

In this study, we identified some traditional medicinal plants, lichens, and mushrooms collected from the Qinghai–Tibetan plateau, China, the extracts of which demonstrated anti-influenza activity for currently relevant influenza viruses, as shown through genetic analysis of the strain A/Novosibirsk/RII-27192S/2020 (H3N2).

These nine original medicinal materials, in addition to medicinal usages, have also served as traditional food for thousands of years in the Qinghai–Tibetan plateau. *A. sativa* and *H. vulgare* are two major cereal crops. The fruits of *H. rhamnoides*, *L. ruthenicum*, and *N. tangutorum* can be eaten raw, or made into juice and fruit snacks. The dendritic part of the roots of lichen *C. rangiferina*, the roots of plant *P. anserine*, and the fruit body of the mushroom *A. luteo-virens* are often eaten as a salad after blanching.

In this study, we tested the antiviral activities of these traditional Qinghai–Tibetan drugs derived from nine extracts of plants, lichens, and mushrooms, and their components, against the influenza virus in MDCK cells. For studying the inhibitory activity of the extracts against the influenza virus, we chose the relevant human influenza virus: the strain A/Novosibirsk/RII-27192S/2020 (H3N2). Currently, this is one of the most relevant strains of the influenza virus, despite the fact that the COVID-19 pandemic has minimized the role of the influenza virus in the overall structure of the incidence of respiratory viral infections [29,30,31].

The analysis of the antiviral activities of the nine extracts revealed that only two extracts, NJBAE and PAWP, had no inhibitory activity on A/H3N2. Notably, NTWP and NJBAE, both from the fruit of *N. tangutorum*, exhibited different anti-influenza effects on the A/H3N2 strain. Therefore, we hypothesize that berry polysaccharides might be more effective at inhibiting A/H3N2 virus than berry anthocyanins, although this assumption needs to be detailed in future studies.

The PAWP extract obtained from the roots of *P. anserina* demonstrated low inhibitory activity against A/H3N2, which may be explained by the low total protein, peptide, and sugar contents. Additionally, the predominant monosaccharide composition of PAWP consists of arabinose, glucose, rhamnose, galactose, and galacturonic acid, representing more than 90% (mol%) of the total carbohydrates, as reported by Xia et al. [32], which might also be related to its low inhibitory activity. Notably, the ALWP extract, obtained from the fruit body of the mushroom, showed excellent antiviral activity, which may be explained by its high total protein and peptide contents compared to the other extracts. According to its chemical composition, ALWP may be composed of some neutral proteoglycans that might produce the anti-influenza effect. However, no reports on the composition of ALWP have been published, and its detailed analysis by our group is still underway. Additionally, the results of our previous study indicated that the LRWP extract, obtained from the fruit of *L. ruthenicum*, and NTWP extract contained rhamnogalacturonan I (RG-I) pectin, whereas HRWP, obtained from the fruit of *H. rhamnoides,* was composed of homogalacturonan (HG) pectin [24,33]. Thus, the differences in the inhibitory activity against the A/H3N2 virus among the extracts suggest that the anti-influenza activity of RG-I pectin may be higher than that of HG pectin. Both extracts obtained from the bran of Gramineae (ASWP (*A. sativa*) and HVWP (*H. vulgare*)), and the extract CRWP-S obtained from the dendritic part of *C. rangiferina*, showed concentration-dependent antiviral activity against A/H3N2. Both ASWP and HVWP contain cereal β-1,3-1,4-D-glucan, which is a soluble dietary fiber with multiple nutritional benefits [34]. Colloidal oat extracts of *A. sativa* exhibit direct antioxidant and anti-inflammatory activities due to the actions of the enzymes monodehydroascorbate reductase and dehydroascorbate reductase [35]. Our previous study showed that CRWP-S might contain several polysaccharides, such as glucans, glucomannans, and galactomannans, and has a protective effect on the alveolar epithelial cells from Pb^2+^-induced oxidative damage [23]. Having virucidal and bactericidal properties, together with low cytotoxicity, makes phytomedicine promising in the fight against viral and bacterial agents.

In our study, the results suggest that extracts of the fruits of *H. rhamnoides*, *L. ruthenicum*, and *N. tangutorum*; the dendric part of *C. rangiferina*; brans of *A. sativa* and *H. vulgare*; and fruit bodies of the mushroom *A. luteo-virens*, exhibit minimal cytotoxicity and strong virucidal properties, indicating their potential as candidates for the development of efficacious and nontoxic drugs against the A/H3N2 virus.

## Figures and Tables

**Figure 1 viruses-14-00360-f001:**
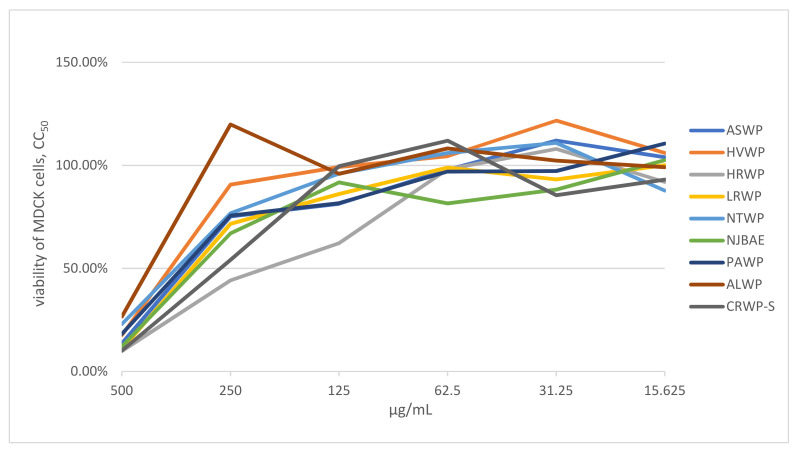
Viability of MDCK cells (%) treated with different concentrations of extracts. Note: CC_50_ represents the concentration of medicinal plant, lichen, or mushroom extract required to reduce the number of viable cells by 50% relative to the control wells without test compound, calculated from dose–response data; extract concentrations are expressed in μg/mL.

**Figure 2 viruses-14-00360-f002:**
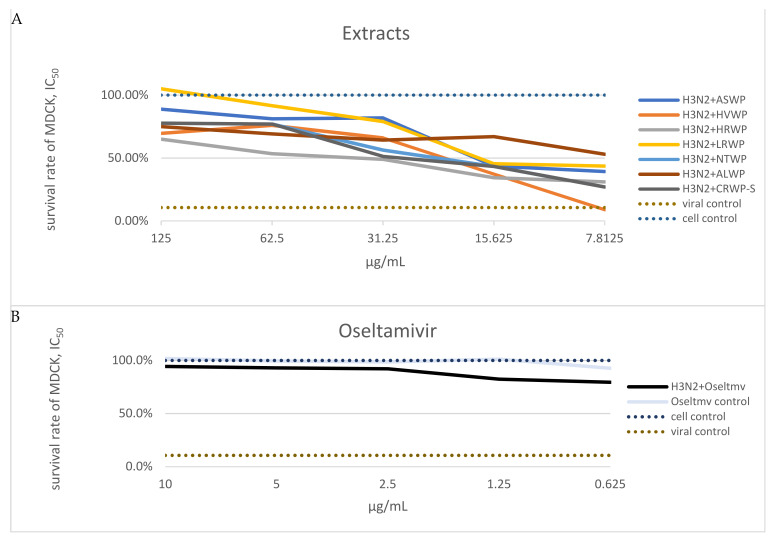
(**A**) Determination of inhibitory concentration values of extracts at which 50% of 100 TCID50 of the A/H3N2 virus was neutralized. Note: Data on extracts NJBAE and PAWP are not shown in the graphs; extract concentrations are expressed in μg/mL. (**B**) Survival rates of MDCK cells: were left untreated (negative control), treated with oseltamivir, treated with oseltamivir and A/H3N2 virus, or treated with the A/H3N2 virus (positive control).

**Table 1 viruses-14-00360-t001:** Origin of extracts and their names.

No.	Origin	Extract Name
1	Bran of *Avena sativa* L.	ASWP
2	Bran of *Hordeum vulgare* Linn. var. nudum Hook. f.	HVWP
3	Fruit of plant *Hippophae rhamnoides* Linn.	HRWP
4	Fruit of plant *Lycium ruthenicum* Murr.	LRWP
5	Fruit of plant *Nitraria tangutorum* Bobr.	NTWP
6	Anthocyanins from plant *Nitraria tangutorum* Bobr. by-products	NJBAE
7	Root of plant *Potentilla anserina* L.	PAWP
8	Dendritic part of lichen *Cladina rangiferina* (L.) Nyl.	CRWP-S
9	Fruit body of mushroom *Armillaria* *luteo*-virens (Aalb.et Schw:Fr) Sacc.	ALWP

**Table 2 viruses-14-00360-t002:** The total carbohydrate, uronic acid, protein and peptide, anthocyanin, and endotoxin contents of the studied extracts.

No.	Extract Name	Total Sugar Content (%)	Uronic Acid Content (%)	Total Protein and Peptide Content (%)	Total Anthocyanin Content (mg CGE per g)	Total Endotoxin Content (EU)/mL	CC_50_ (μg/mL)	IC_50_ (μg/mL)
1	ASWP	97.42	n.d.	n.d.	n.d.	<0.5	395.46	19.53 ± 0.41
2	HVWP	93.20	n.d.	n.d.	n.d.	<0.5	354.56	23.73 ± 0.62
3	HRWP	94.73	61.93	1.32	0.24	<0.5	186.36	36.46 ± 1.12
4	LRWP	95.37	3.32	0.01	5.53	<0.5	409.11	17.58 ± 0.38
5	NTWP	97.25	2.78	2.10	6.12	<0.5	372.74	23.80 ± 0.38
6	NJBAE	5.52	n.d.	n.d.	723.60	<0.5	286.37	n.d.
7	PAWP	69.35	9.51	0.07	6.79	<0.5	386.38	n.d.
8	ALWP	51.43	0.08	10.22	7.33	<0.5	440.90	7.81 ± 0.12
9	CRWP-S	92.16	0.01	2.92	n.d.	<0.5	277.28	31.25 ± 2.51

Note: CGE, cyanidin-3-glucoside equivalents; EU, endotoxin units; CC_50_, the cytotoxic concentration of the extracts that causes death to 50% of viable cells; IC_50_, inhibitory concentration of extract at which 50% of A/H3N2 virus was neutralized; n.d., not defined.

## Data Availability

Not applicable.

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
