# Peer review of "Anti-Influenza Activity of Medicinal Material Extracts from Qinghai–Tibet Plateau"

_viruses, 2022, doi:10.3390/v14020360_

Round 1

Reviewer 1 Report

In this manuscript, the authors examined the effect of 9 extracts from 8 plants and one mushroom on influenza A virus infection in MDCK cells. The 8 plants and one mushroom are traditionally used as indigenous medicines in some areas in China. They found that most of the tested extracts protected the infected MDCK cells from cell death compared to non-treated, infected cells. The results are interesting and, if proven, can serve as scientific evidence for the antiviral efficacy against influenza viruses of those traditional medicinal plants/mushrooms. However, the results are preliminary and some critical experiments are missing. For example, to support the notion that those plant/mushroom extracts have anti-viral effect, virus replication (virus titers) in the cells treated with different concentration of the extracts should be performed. Do these plant/mushroom extracts have any effect on the replication of other influenza A virus strains? In addition, if the authors think that certain proteins/peptides and sugars play a major role in inhibiting influenza virus infection, they should include control in the tests (e.g., will the virus replication in MDCK cells be affected by equivalent levels of other proteins/peptides (e.g., BSA) and sugars (e.g., sucrose)?).   

Author Response

Dear Reviewer #1,

Thank you for your useful comments and questions which helped improve the manuscript. Please see below each point for the detailed response on how we incorporated your feedback into the manuscript.

  1. The results are interesting and, if proven, can serve as scientific evidence for the antiviral efficacy against influenza viruses of those traditional medicinal plants/mushrooms. However, the results are preliminary and some critical experiments are missing. For example, to support the notion that those plant/mushroom extracts have anti-viral effect, virus replication (virus titers) in the cells treated with different concentration of the extracts should be performed.

Response. In this study, cytotoxicity assessments were carried out on MDCK cell culture, in which different doses of extracts were studied. An IC50 assessment was also carried out that revealed viral replication of A/H3N2 in MDCK cells was considered with different concentrations of the extracts. Thus, we lay out that in currently research some of critical experiments such are determination of CC50 and IC50 already done.

  1. Do these plant/mushroom extracts have any effect on the replication of other influenza A virus strains?

Response. We agree with the Reviewer that it is a very interesting question. But in this study, we did not set ourselves the goal of conduct research on other influenza viruses. The A/H3N2 was chosen based on the fact that it is constantly circulating and one of the most relevant influenza pathogen.

Thank you for your recommendation, we will try to pay more attention to this issue in future studies!

  1. In addition, if the authors think that certain proteins/peptides and sugars play a major role in inhibiting influenza virus infection, they should include control in the tests (e.g., will the virus replication in MDCK cells be affected by equivalent levels of other proteins/peptides (e.g., BSA) and sugars (e.g., sucrose)?).

Response.

The composition of the virus growth medium includes BSA and glucose, which is necessary for cell growth, but does not affect on the reproduction of the virus and does not have anti-influenza properties. That is why we didn`t include in the test equivalent levels of other proteins/peptides (e.g., BSA) and sugars (e.g., sucrose). We assume that extracts with the predominantly composition monosaccharides (arabinose, glucose, rhamnose, galactose, and galacturonic acid), as it was mentioned by Xia et al. [34], may demonstrated low inhibitory activity against A/H3N2 virus. These results lead to further investigation about characterization of active compounds and their specific mechanism against influenza virus.

Also the Manuscript was corrected errors in English writing according to recommendation.

Thank you for your recommendation, we will try to pay more attention to this issue in future studies!

Reviewer 2 Report

Here, the authors describe the extraction and isolation of compounds from a variety of plants and their ability to block a clinical isolate of H3N2 from binding to the MDCK cells.

The statement in the abstract that two polysaccharides from a N. tangutorum show anti-viral activity suggests that berry polysaccharides might be more effective than anthocyanins does not have enough data to be supported.

The paper is generally interesting but would need to be improved by a couple experiments to improve the understanding of the compounds which are being used and the effects that they are having on the MDCK cells.

One is HPLC of the purified compounds to understand the distribution of the polysaccharides and anthocyanin. This may also help in narrowing down active compounds.

The second is an assay to detect the amount of productive infection that is happening in the presence of these compounds associated with figure 2. This would answer important questions about whether the compounds reduce viral output, sensitize cells to death during infection or block viral life cycle.

Additionally, it would be useful to test survivability of a few different cell lines with these compounds to show that their cell death properties are consistent.

Author Response

Dear Reviewer #2,

Thank you for your useful comments and questions which helped improve the manuscript. Please see below each point for the detailed response on how we incorporated your feedback into the manuscript.

  1. The statement in the abstract that two polysaccharides from a N. tangutorum show anti-viral activity suggests that berry polysaccharides might be more effective than anthocyanins does not have enough data to be supported. The second is an assay to detect the amount of productive infection that is happening in the presence of these compounds associated with figure 2. This would answer important questions about whether the compounds reduce viral output, sensitize cells to death during infection or block viral life cycle.

Response. In the current research we conducted a general screening of extracts, but a more detailed elaboration of the mechanisms of action of compounds requires further research. Our obtained results may to lead to further investigation about characterization of active compounds and their specific mechanism against influenza virus. This plant/mushroom extracts holds promising antiviral effects and may serve as novel amply available anti-influenza drugs. A more thorough view on the active compounds is required to understand the impact of molar mass, molar ratio, solubility and viscosity of the extracts compounds, that is also help to narrowing down the component that enough to preventing the development of viral infection.

We agree with the Reviewer that the phrase «So, we deduced that berry polysaccharides might be more effective at inhibiting A/H3N2 virus than berry anthocyanins” in the abstract should be changed. We toned down it into “So, we may hypotesized that berry polysaccharides might be more effective at inhibiting A/H3N2 virus than berry anthocyanins, but this assumption needs to be detailed in future studies.”, and put this phrase into the Discussion part.

  1. The paper is generally interesting but would need to be improved by a couple experiments to improve the understanding of the compounds which are being used and the effects that they are having on the MDCK cells.

Response. We used the MDCK cells only because they are widely used and are the ‘gold standard’ in studies of influenza viral activity. Thank you for your recommendation, we will try to pay more attention to this issue in future studies!

Also the Manuscript was corrected errors in English writing according to recommendation.

Finally, we corrected the text and try to fix all the comments. Please see it again.

Round 2

Reviewer 1 Report

In the revised version of this manuscript, English is significantly improved. However, the authors did not meaningfully address the issues I raised previously.

Author Response

Dear Reviewer #1,

Thank you for your useful comments and questions which helped to improve the Text!

In this Manuscript we purposed to identify of some traditional extracts of medicinal plants, lichens, and mushrooms (Avena sativa L., Hordeum vulgare Linn. var. nudum Hook. f., Hippophae rhamnoides Linn., Lycium ruthenicum Murr., Nitraria tangutorum Bobr., Nitraria tangutorum Bobr. by-products, Potentilla anserina L., Cladina rangiferina (L.) Nyl., and Armillaria luteo-virens) collected from the Qinghai–Tibetan plateau, China, which may demonstrate anti-influenza activity for currently relevant influenza A/H3N2 virus. We did not set the goal to search the certain proteins/peptides and sugars that may play a major/minor role in inhibiting influenza virus infection. Our study on the complex action of selected extracts against influenza A/H3N2 virus. However, some studies on the properties of the extracts were carried out in our previous studies, from which such properties as antioxidant potential of polysaccharides of lichen (C. rangiferina) and some medicinal plants [22], or antifatigue activity of their water-soluble polysaccharides [16]. In addition, we continue to study the structural analysis of extracts [25, 35], which, as we assume, will allow to identify the specific component responsible for antiviral action in the future. Additionally, we completed the сontrol experiment with Oseltamivir that in turn is the ‘gold standard’ in studies of anti-influenza drugs that prevent and treat diseases. Please, kindly find the added information in the Text that highlighted in yellow and also included in the Figure 2(B).

We believe that this manuscript is appropriate for publication in Special Issue "State-of-the-Art Respiratory Viruses Research in Russia" of `Viruses` because it describe the screening analysis of nine extracts from eight medicinal plants and one mushroom that exhibit minimal cytotoxicity on MDCK and strong virucidal properties against the A/H3N2 virus. This information is very interesting and important for community involved.

Reviewer 2 Report

Thank you for your thoughtful responses

Author Response

Thank you for taking the time and effort necessary to review our Manuscript! We sincerely appreciate all valuable comments and suggestions!